

# Effects of alkali treatment on the mechanical and thermal properties of sisal/cattail polyester commingled composites

Silas Mogaka Mbeche[1,2] and Timothy Omara[2,3,4]

[1] Department of Manufacturing, Industrial and Textile Engineering, School of Engineering, Moi University, Uasin Gishu County, Eldoret, Kenya

[2] Africa Centre of Excellence II in Phytochemicals, Textiles and Renewable Energy (ACE II PTRE), Moi University, Uasin Gishu County, Eldoret, Kenya

[3] Department of Chemistry and Biochemistry, School of Biological and Physical Sciences, Moi University, Uasin Gishu County, Eldoret, Kenya

[4] Department of Quality Control and Quality Assurance, Product Development Directory, AgroWays Uganda Limited, Jinja, Uganda

Corresponding authors
Silas Mogaka Mbeche, silambeche07@gmail.com
Timothy Omara, prof.timo2018@gmail.com, prof.timo2018@mu.ac.ke

## ABSTRACT

Environmental and energy conservation pressure has led to a dramatic increase in the need for economically feasible lightweight materials that can be better substitutes for non-biodegradable materials in reinforced composites. In this study, the mechanical and thermal properties of polyester resin composites hybridized with a blend of untreated and alkali treated sisal (*Agave sisalana*) and cattail (*Typha angustifolia*) fibers were evaluated. Composites were fabricated by a hand lay-up technique at an optimal hybrid fiber weight fraction of 20 wt% and a constant sisal/cattail fiber blend ratio of 75/25. Flexural, tensile, compressive and impact strengths and moduli, as well as thermal conductivity of the composites, were evaluated following ASTM and ISO test methods. Analytical results indicated that alkali pre-treatment of the fibers enhanced the mechanical properties of the hybrid polyester composites though only marginal differences were recorded in the thermal conductivity of the composites fabricated with treated and untreated fiber blends. Morphological examination revealed that the major failure modes were fiber pull-outs and fiber fracture in composites fabricated with untreated and treated fiber blends, respectively. The composites produced could find non-structural applications as ceiling boards, electronic and food packaging materials but their properties such as wettability, crystallinity, flammability and other thermal properties need to be further investigated.

## INTRODUCTION

Reinforcement of construction materials such as mud bricks using natural fibers (grass and straw) have been witnessed since the beginning of the era of human civilization (*Rizal et al., 2018*). With the industry 4.0 revolution, natural fibers from sisal, cattail, flax, date palm, coir, oil palm empty fruit bunch, ramie, sugar palm, bamboo, hemp, rubber wood, jute,

pineapple leaf and kenaf have been subject of obsessive research for their possible use in reinforcement of composite materials (*Bongomin et al., 2020*; *Gupta et al., 2016*; *Khanam et al., 2007*; *Nasir et al., 2014*; *Venkateshwaran et al., 2011*). This is primarily because they possess superior properties such as high specific strength and stiffness, low density and toxicity, carbon dioxide neutrality, are biodegradable and are readily available. This make them good substitutes for expensive glass, carbon and aramid when used in composites for low load bearing and thermal applications (*Asim et al., 2017*; *Malkapuram, Kumar & Negi, 2009*; *Pickering, Efendy & Le, 2016*; *Sanjay, Arpitha & Yogesha, 2015*; *Sliseris, Yian & Kasal, 2016*). Moreover, their inclusion in composites economizes the volume of the polymeric matrix consumed, conferring additional logistic advantages (*Frollini et al., 2013*). The salient drawbacks with natural fibers when commingled in composites is their relatively high moisture absorption tendency and poor compatibility with the matrix (*Gupta et al., 2016*; *Venkateshwaran et al., 2011*). These negative characteristics are primarily due to their inherent heterogenous content such as carbohydrates (cellulose, hemicellulose, starch), lignin, pectin, waxes, fats and other polar compounds (*Mortazavi & Moghaddam, 2010*; *Sun & Cheng, 2002*). Thus, weak interface adhesion between natural fibers and polymer matrices are due to the differences in the wettability of natural fibers (inherently hydrophilic) and the polymer matrix (usually hydrophobic). This phenomenon reduces the efficiency of stress distribution from the matrix to the fiber and simultaneously diminishes the mechanical properties of such natural fiber-reinforced composites (*Punyamurthy et al., 2012*). So often, alkali treatment (typically with sodium hydroxide solution) or treatment with coupling agents are preferred for reworking the properties of natural fibers to improve interfacial bonding between the fibers and the matrices in the resultant composites (*Aziz & Ansell, 2004*; *Boopathi, Sampath & Mylsamy, 2012*; *El-Abbassi et al., 2015*; *Manalo et al., 2015*; *Reddy et al., 2013*; *Rout et al., 2001*; *Verma et al., 2013*; *Goud & Rao, 2011*).

Both sisal (*Agave sisalana*) and cattail (*Typha angustifolia*) are readily available plants in Kenya, the latter being a wild marginal weed (*Colbers et al., 2017*; *Committee on Commodity Problems, 2017*; *Mukherjee & Satyanarayana, 1984*; *Phologolo et al., 2012*). Sisal fibers are widely used due to their availability with each plant producing 200–250 leaves and each leaf producing 1,000–1,200 fiber bundles. Thus, a normal leaf weighing about 600 g yields about 3% by weight of fibers (*Mukherjee & Satyanarayana, 1984*). Due to its better mechanical properties and abundance in most parts of Kenya, sisal fibers could be a promising reinforcement material in hybrid composites. On the other hand, fibers from cattail plant leaves are identical to hemp (jute) fibers and thus can be similarly utilized in textile and composite applications (*Baldwin & Cannon, 2007*; *Mortazavi & Moghaddam, 2010*; *Ramanaiah, Prasad & Reddy, 2011*). The use of cattail fibers could control the invasive cattail weed, generate employment opportunities while conserving the environment.

Investigation of the properties of sisal and cattail fibers when used singly or in combination, with other fibers or materials in composites have been done by preceding researchers. *Joseph et al. (2003)* evaluated the dynamic mechanical properties of polypropylene composites reinforced with treated and untreated short sisal fibers. They examined the resultant composites with reference to fiber loading, fiber length, chemical treatment, frequency and temperature and deduced that inclusion of sisal fibers increased

the storage and loss moduli of the composites as the reinforcement imparted by the fibers allowed stress transfer from the matrix to the fiber. A fiber length of 2 mm was cited by the team as necessary for attaining maximum dynamic and loss moduli. Another study by *Gupta et al. (2016)* which evaluated the mechanical properties of alkali treated sisal/hemp fiber reinforced hybrid epoxy composites indicated that increase in tensile and flexural strengths were registered at 40 wt% of sisal/hemp fiber blend, and increase in the weight percentage of fiber blend increased the hardness strength of the composites. In another such concerted investigation, *Bichang'a, Wambua & Nganyi (2017)* evaluated the effect of alkali treatment on the mechanical properties of a woven sisal fabric reinforced epoxy composite fabricated at 40% fiber weight fraction. The team inferred that chemical treatment of sisal fabric with 4% (w/v) sodium hydroxide solution for 1 h at room temperature improved the mechanical properties of the resultant composite. Further, *Samuel, Agbo & Adekanye (2012)* disclosed that the mechanical properties of ukam and sisal fiber reinforced composites were greatly influenced by alkali treatment of the fibers.

For cattail plant, a patent for using its parts for production of thermal insulation materials was filed in 1962 (*Google Patents, 2019*) and several others have been filed with promising results from feasibility studies indicating its possible use in the manufacture of insulation plates and blow-in insulation boards (*Colbers et al., 2017*; *Naporo, 2012*). In an investigation by *Rizal et al. (2018)*, it was reported that treatment of cattail (*Typha* species) with 5% (w/v) sodium hydroxide solution improved the interfacial shear strength of hybrid epoxy composites from 2.240 MPa to 2.718, 3.753, 3.960 and 4.185 Mpa after 1, 2, 3 and 4 h of treatment respectively. Composite specimens after 4 h of alkali treatment had the highest tensile strength of 37.4 MPa compared to 29.2 Mpa in untreated *Typha* reinforced epoxy composites. *Rizal et al. (2019)* inferred that the mechanical properties and crystallinity index of composites reinforced with cattail fibers previously treated with 5% (w/v) sodium hydroxide solution increased with processing time. The current study investigates the effect of alkali treatment on the mechanical and thermal properties of sisal/cattail reinforced polyester composites.

# MATERIALS & METHODS

## Fiber samples and chemicals used

Sisal fibers, traditionally popular for making twine and ropes, were generously supplied by Lomolo Sisal Estate Ltd, Mogotio, Baringo county, Kenya. Green mature cattail plant leaves were obtained from cattail (*Typha angustifolia*) plants from a swamp in the propinquity of Moi University staff quarters, Uasin Gishu county, Eldoret, Kenya. Analytical reagents: unsaturated polyester resin (UPR), methyl ethyl ketone peroxide (MEKP) and acetic acid, double distilled water and Sodium hydroxide pearls (extra pure) were supplied by Henkel Chemicals (E.A.) Ltd, Industrial area, Nairobi, Kenya and Moi University Textile Laboratory, Uasin Gishu county, Eldoret, Kenya respectively. Mechanical properties of the neat unsaturated polyester resin (UPR, GP 1778) were density-1.23 $g/cm^3$, tensile strength-29.20 MPa, tensile modulus- 2,194.70 MPa, flexural strength- 70.00 MPa, impact strength- 9.00 $kJ/m^2$ and an elongation at break of 4.20%.

## Preparation of sisal and cattail fibers

Cattail leaves were separated from the stalk grouping at the leaf base followed by mechanical decortication process to extract the fibers. Fibers were subsequently dried at 80 °C in an oven to constant weight to eliminate excess moisture that would otherwise result in poor fiber-matrix adhesion. Sisal fibers as supplied were cleaned with warm distilled water (for those not to be alkali treated) to remove chlorophyll, leaf juices, adhesive solids and soluble impurities after which they were dried.

## Fiber surface treatment

Fiber surface modification was achieved by submersing sisal and cattail fibers in 4% and 5% (w/v) sodium hydroxide solution at room temperature for 1 h respectively (*Bichang'a & Ayub, 2017*; *Dedeepya, Raju & Kumar, 2012*; *Rizal et al., 2019*). After treatment, the fibers were rinsed thoroughly with distilled water acidified with acetic acid (1% w/v) to neutralize excess sodium hydroxide in the fibers. The pH of the rinses was monitored using Hanna 211 digital microprocessor-based bench top pH/mV/°C meter (Hanna instruments, Italy) previously calibrated using pH 4.01, 7.01, 10 buffers. The pH electrode was thoroughly rinsed with distilled water in between different measurements. The rinsed fibers were subsequently dried.

## Characterization of sisal and cattail fibers

Both treated and untreated sisal and cattail fibers (pre-dried in an oven for 1 h at 80 °C to remove excess moisture that could lead to poor fiber-matrix adhesion) were characterized by determining their linear densities and tensile properties (tenancies).

Linear densities of the fibers were determined as per ASTM D1577-2001 by weighing out known lengths of the fibers. Thirty (30) fibers, each from treated and untreated sisal and cattail fibers were picked randomly and then cut to a length of 300 mm (as per the universal tensile testing machine gauge length and the manufacturer's instructions) to form four (4) bundles of fibers. Each of the four test specimen bundles were separately weighed. From the measured fiber weights and the number of fiber specimen in each bundle, the weight of each fiber in the four bundles were determined. With these weights in grams, linear density in tex was determined by dividing the fiber weight by its length in kilometers.

Tensile strength of the fibers were determined as per ASTM D3822M-2001 under ambient conditions using a universal tensile testing machine (UTM-TH2730, Rycobel, Belgium) at a gauge length of 300 mm and speed of five mm/min. Tensile strength was determined for the four bundles of treated and untreated fibers by taking an average of 30 tests for each. From these tests, fiber tensile strength in terms of breaking tenacity (cN/tex) were determined by dividing the breaking force (cN) by the linear density (tex) of the treated and untreated sisal and cattail fibers. The assumptions made were that the fibers are cylindrical in shape.

## Composite fabrication

Sisal/cattail fiber reinforced polyester composites were prepared by simple hand (a wet) lay-up technique as described by *Borah et al. (2016)* with slight modifications. A mould measuring $310 \times 310 \times 25$ mm was fabricated using a polished iron metal sheet from which

composites of dimensions 300× 300 mm were prepared. The mould was cleaned using acetone, followed by application of mould release agent (MR8) on the inner surfaces. The inner surfaces were then covered with aluminium foil to avoid the chances of composites sticking onto the mould surface and to provide good surface finish. The experimental design for the amount of matrix material and the reinforcements used in the composites follows from a preceding study (*Mbeche, Wambua & Githinji, 2020*) which indicated that the optimal weight fraction was 20% with 75/25 sisal/cattail fiber blend for optimal mechanical properties of the resultant polyester composites.

Unsaturated polyester resin (UPR) and hardener (MEKP) were mixed in a ratio of 1:0.02 by mass as per the manufacturer's instructions and stirred thoroughly. The resin was mixed with blended fibers and stirred for 15 min to ensure uniform dispersion of fibers within the resin. The content was then poured into the mould and then spread gently to ensure uniform thickness of the resultant composite. To prevent air entrapment during fabrication, a thin plastic sheet (velvex) was used to cover the mould and then pressed gently and uniformly using a pressure roller. The composites were allowed to cure at ambient conditions for 6 h under 3.27 kN/m$^2$ compressive pressure after which they were trimmed prior to mechanical tests. Analytical weighings were done using a calibrated Mettler PM200 digital analytical balance (Marshall Scientific, Hampton, NH, USA).

## Evaluation of mechanical properties and fractography studies

Composite samples for various mechanical tests were conditioned for 48 h at ambient conditions of temperature ($23 \pm 2$ °C) and relative humidity (65%) prior to evaluation at the Materials Engineering Laboratory of Multimedia University, Nairobi, Kenya.

Three-point flexure tests were conducted in accordance with ASTM D790-2003 standard using a universal material testing machine (Model UT-10, Enkay Enterprises, India) at a loading rate of 2 mm/min at Rivatex East Africa Limited Textile laboratory. Tensile, flexural and compressive moduli were computed from the stress–strain curves. Impact strength was estimated using a Charpy impact tester (Model HLE, Enkay Enterprises, India) as per ISO 179-1:2000 standard.

Tensile and compression tests were conducted using a universal testing machine (UTM-TH2730, Rycobel, Belgium) with a maximum load cell of 5 kN. The tensile and compressive properties were determined in accordance with ASTM D638-2014 and ASTM D3410M-2003 standards at loading rates of 2 mm/min and 5 mm/min respectively.

Surface morphology of untreated and treated sisal/cattail polyester hybrid composites were investigated using MSX-500Di Scopeman Digital Microscope (Herter Instruments, Barcelona, Spain).

## Evaluation of thermal conductivity

Thermal conductivity tests were done using a thermal conductivity apparatus (Model P5687, Cussons Technology, UK) in the Thermodynamic Engineering Laboratory of Jomo Kenyatta University of Agriculture and Technology, Nairobi, Kenya in accordance with ASTM C518-1998.

## Analytical quality assurance and quality control

All reagents used were of high analytical purity. Equipment such as pH meter and analytical balance were calibrated prior to use. All samples were analyzed at least in triplicate to obtain a relative uncertainty of less than 5% (*Omara et al., 2019*).

## Statistical analysis of results

Analytical data were captured in Microsoft Excel 2016 (Microsoft Corporation, USA) for preliminary analysis. Data were checked for normality prior to statistical evaluation using the Kolmogorov–Smirnov test and subsequently presented as means of quintuples with errors as standard deviations attached. Paired $t$ test was performed to identify any significant differences between groups. All analyses were performed at a 95% confidence interval (with differences in mean values accepted as being significant at $p < 0.05$) using Sigma Plot statistical software (v14.0, Systat Software Inc., San Jose, CA, USA) (*Omara et al., 2019*).

# RESULTS & DISCUSSION

## Properties of sisal and cattail fibers

Linear density (in tex) and tenacity (in cN/tex) were used to characterize the fibers. Higher mean linear densities (26.17 ± 8.33 tex and 35.17 ± 54.96 tex) were recorded for untreated sisal and cattail fibers as compared to 10 ± 5.07 tex and 12.33 ± 5.42 tex recorded in the alkali treated fibers. On the other hand, the mean tenacity of treated sisal and cattail fibers were 146.26 cN/tex and 35.35 cN/tex compared to 23.52 cN/tex and 9.46 cN/tex recorded in the untreated fibers respectively (Table S1). These differences could be attributed to the reduction in fibre diameter due to the loss of weight resulting from the removal of carbonaceous materials after alkali treatment (Fig. 1) (*Gañan et al., 2005*; *Ikramullah et al., 2019*; *Reddy et al., 2013*). The micrographs (Fig. 1) confirm that there were reductions in the diameter of sisal and cattail fibers after alkali treatment for 1 h. The high tenacity values of alkali treated sisal fibers can be attributed to the removal of lignin and other soluble impurities thereby increasing the aspect ratio and thus tenacity of the fibers (*Mahato, Goswami & Ambarkar, 2014*). Paired $t$ test indicated that there were significant differences ($p < 0.05$) between the tenacity of treated and untreated sisal and cattail fibers. A comparable value of 7.83 tex for treated sisal fibres was reported by *Mahato, Goswami & Ambarkar (2014)*. *Rezig, Jaouadi & Msahli (2014)* reported a linear density of 32 tex for untreated cattail fibers and a linear density of between 10 to 30 tex for alkali treated cattail fibers. *Rezig et al. (2016)* recorded a tenacity of 12.41 cN/tex in an optimized cattail fibre extraction process from cattail plant leaves using 20 g/L of sodium hydroxide at 100 °C. *Mortazavi & Moghadam (2009)* and *Mortazavi & Moghaddam (2010)* reported tenacities of 30.17 ± 4.7 cN/tex and 34.87 cN/tex for untreated cattail fibres and those treated with 6% sodium hydroxide in 3% ethylenediaminetetraacetic acid (EDTA).

## Effect of alkali treatment on flexural, tensile and compressive strengths of the polyester hybrid composites

Figure 2 illustrates the effect of alkali treatment on flexural, tensile and compressive strengths of the composites at 20 wt% hybrid fiber weight fraction and 75/25 sisal/cattail

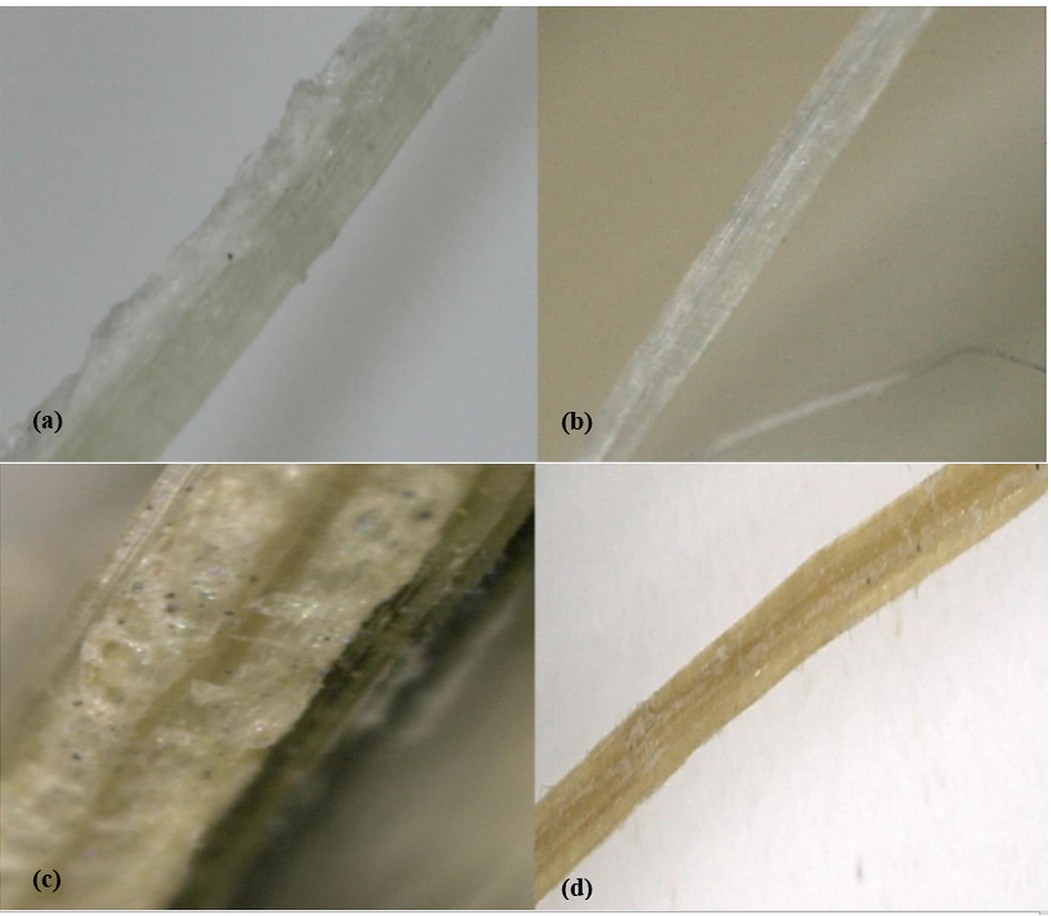

**Figure 1** **Surface imaging of the fibers.** (A) Untreated sisal, (B) alkali treated sisal, (C) untreated cattail, and (D) alkali treated cattail (All micrographs taken at x115).

fiber in the hybrid composites. As illustrated, hybrid composites prepared from a blend of treated sisal and cattail fiber had better mechanical properties than those from untreated fibers. The flexural, tensile and compressive strengths improved by 24.64%, 19.56% and 18.70% respectively with alkali treatment. Paired $t$ test results revealed that there were significant differences ($p < 0.05$) between the mean values of the evaluated mechanical properties of composites reinforced with untreated and treated fiber blends. High-resolution fracture behavior images after tensile testing of the composites (Figs. 3A–3B) depicts that there were fiber pull outs from the matrix in untreated hybrid composites signifying that there was poor adhesion between the fibers and the matrix (*Asim et al., 2017*). Cattail fiber-pull outs were observed, and this could be due to the lower strength of cattail fibers which resulted in their breakage as compared to sisal fibers.

Cellulose, hemicellulose and lignin are the main components of natural fibers. Whereas cellulose is the principal component in natural fibers, hemicellulose on the other hand is a cementing matrix between cellulose and lignin that confer rigidity to plants. Further, cellulose (a semi crystalline polysaccharide) and hemicellulose are hydrophilic, while lignin

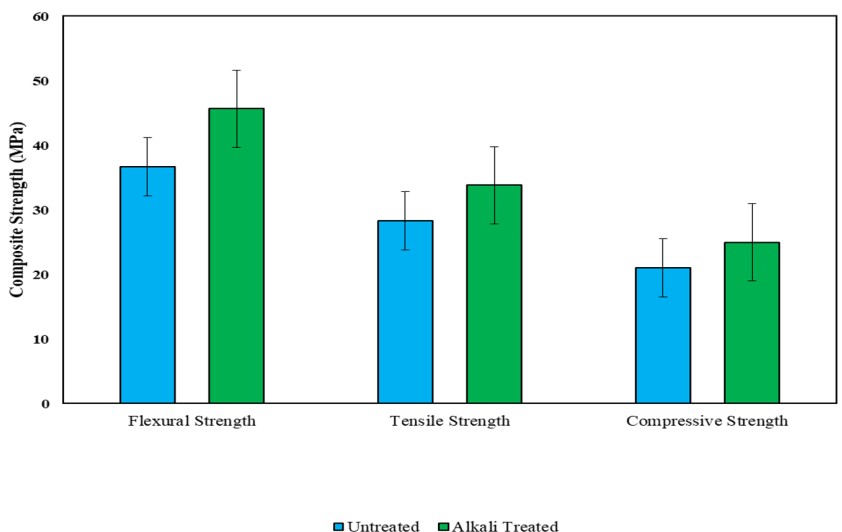

**Figure 2** **Effect of alkali treatment on the flexural, tensile and compressive strengths of the hybrid composites.**

is relatively hydrophobic (*Zhou, Fan & Chen, 2016*). Interface bonding in fiber reinforced composites is through electrostatic, chemical bonding and mechanical interlocking mechanisms (*Matthews & Rawlings, 1999*). The latter is dominant when fibers surfaces are rough. This mechanism increases shear strength of the fiber-matrix interface. On the other hand, chances are that different types of bonding between the fiber and the matrix interfaces may occur and act synergistically (*Pickering, Efendy & Le, 2016*). Thus, alkali treatment in this study might have additionally interfered with hydrogen bonds in the chemical structure of the fibers, increasing fiber surface roughness. Further, alkali treatment increases crystallinity index of fibers, enhancing formation of hydrogen bonds between cellulose chains and hence chemical bonding between the fibers in composites (*Gassan & Bledzki, 1999*; *Mylsamy & Rajendran, 2011a*; *Sreekala, Kumaran & Thomas, 2001*). The surface roughness of natural fibers interestingly increases with increase in the duration of alkali treatments (*Sangappa et al., 2014*). However, it should be over emphasized that treatment at high alkali concentrations or subjection to long alkali treatment periods which have been avoided in this study often lead to significant reduction in the mechanical performance of the treated fibers (*Jacob, Thomas & Varughese, 2004*; *Mishra et al., 2003*; *Rizal et al., 2018*).

Therefore, in hybrid composites with treated fibers (Figs. 3C–3D), there were fewer fiber pull-outs but more fiber fracture and twisting at the broken ends indicating a relatively strong bond between the fibers and the matrix. Higher breakages of cattail fibers were also observed in treated hybrid composites, and this could be due to their low strengths as compared to sisal fibers. The lower strength of cattail fibers in this study may be attributed to the extraction process used. Cattail fibers were manually decorticated from their leaves which could have resulted in some breakages and thus compromising their strength. Furthermore, high cattail fiber breakages in treated hybrid composites may be attributed

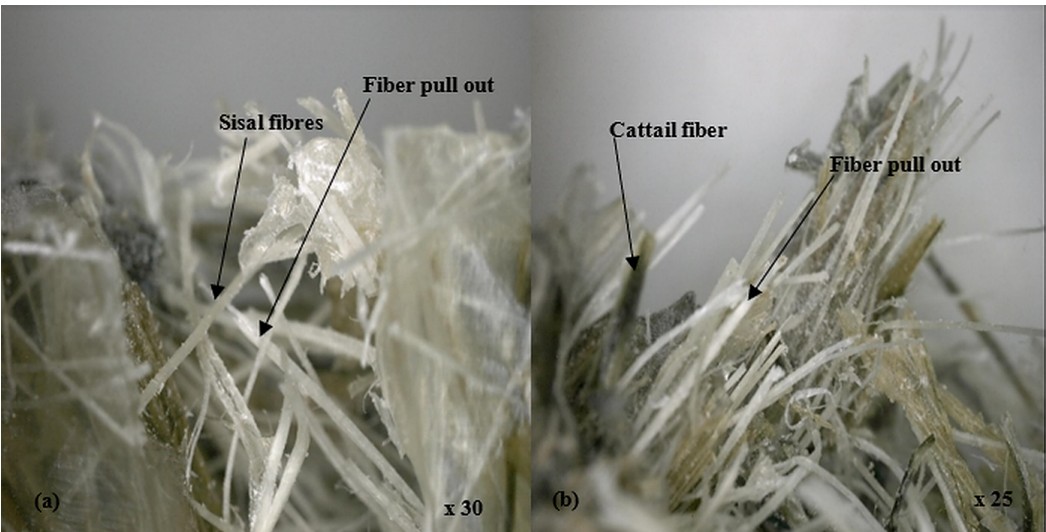

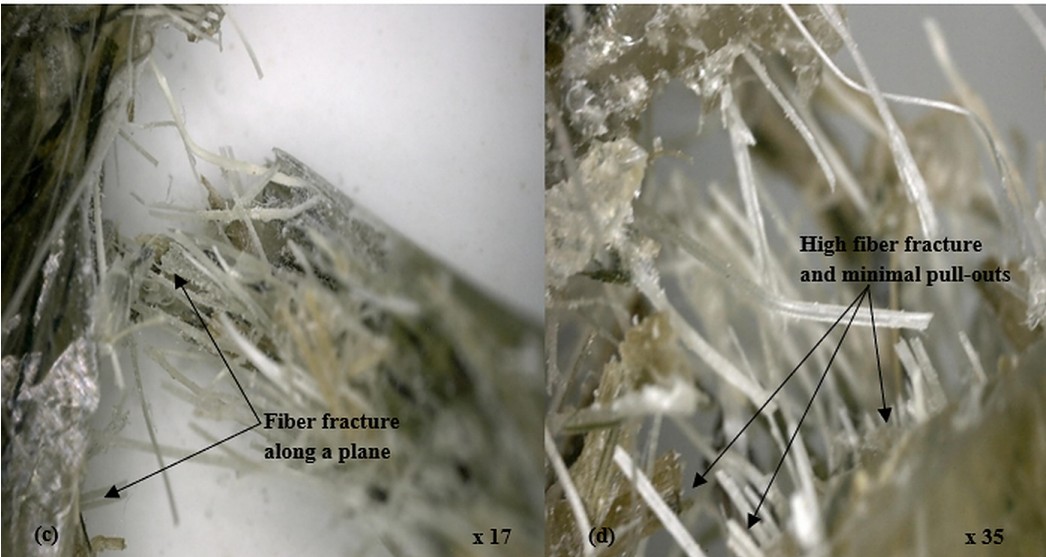

**Figure 3 Micrographs of polyester hybrid composites showing the fracture surfaces in the composites.** (A–B) Composites hybridized with untreated sisal/cattail fibers, and (C–D) composites hybridized with treated sisal/cattail fibers.

to their poor alkali treatment, since most of these extracted fibers were in bundles leading to poor impregnation of sodium hydroxide during treatment. Improvement in mechanical properties following alkali treatment can be attributed to better fiber/matrix interface, due to changes in surface topography of the fibers, leading to increased mechanical interactions with the matrix as well as increasing fiber wettability by removal of some of the cementing components (lignin, pectin, hemicellulose, fats) on the fiber surfaces (*Bichang'a, Wambua & Nganyi, 2017*; *Gañan et al., 2005*; *Mwaikambo & Ansell, 1999*; *Rizal et al., 2018*). This trend is comparable to that reported by *Alavudeen et al. (2011)* in which alkali treatment of randomly mixed banana/kenaf hybrid polyester composites increased the tensile strength

of the composites by 14.1% from 31.9 MPa to 36.4 MPa at 50 wt%. *Senthilkumar & Ravi (2017)* also reported an improvement in flexural strength by 47.78% from 82.74 MPa to 122.27 MPa for untreated and 6% sodium hydroxide sisal fiber treated composites respectively in hybrid epoxy composites. *Rizal et al. (2018)* got similar findings in which alkali treatment of cattail fibers recorded flexural strengths of 44.50 MPa and 69.50, 77.20, 50.30 and 49.80 MPa for epoxy composites with untreated and treated cattail fibers for 1, 2, 4 and 8 h respectively. The highest tensile strength recorded was 37.40 MPa for fibers treated for 4 h, while composites with untreated fibers had tensile strength of 29.20 MPa.

## Effects of alkali treatment on flexural, tensile, compressive moduli of the polyester hybrid composites

The effect of alkali treatment on flexural, tensile, compressive moduli of sisal/cattail hybrid polyester reinforced composites at a constant hybrid fiber weight fraction of 20 wt% and sisal/cattail fiber blend of 75/25 in the hybrid composites are shown in Fig. 4. It is evident that treatment of sisal and cattail fibers resulted in an increase in flexural, tensile and compressive moduli of the hybrid composites. At 20 wt% fraction and 75/25 sisal/cattail fiber content in the hybrid, flexural, tensile and compressive moduli increased by 10.44%, 12.97% and 17.26% respectively. Paired $t$ test indicated that there were significant differences ($p <$ 0.05) in the means of flexural, tensile and compressive moduli of the treated and untreated hybrid composites. The observed trend, therefore, is a clear indication that alkali treatment of sisal and cattail fiber improved the flexural, tensile and compressive moduli of the hybrid polyester composites. This could be because alkali treatment increases surface roughness (exposes hydroxyl groups) of the fibers to the matrix due to the removal of lignin and other impurities from the fiber surface, thereby improving fiber-matrix adhesion (*Ikramullah et al., 2019*; *Mortazavi & Moghaddam, 2010*). A comparable pattern of the effect of alkali treatment on flexural, tensile and compressive moduli was reported by *Bichang'a, Wambua & Nganyi (2017)* for woven sisal reinforced epoxy composites in which 12.97%, 31.19% and 34.98% increments in the flexural, tensile and compressive moduli of the composites respectively were registered. *Rizal et al. (2018)* also noted a similar enhancement of tensile modulus of epoxy composites reinforced with alkali treated cattail fibers.

## Effects of alkali treatment on impact strength of the hybrid polyester composites

Impact strength performance of natural fiber reinforced composites is influenced by factors such as fiber and matrix properties, as well as fiber and matrix interface properties (*Rizal et al., 2018*). In this study, the average impact strength of the composites reinforced with untreated fibers was $23.19 \pm 0.10$ kJ/m$^2$. Thus, it improved with alkali treatment of sisal and cattail fibers by 16.73% to an average maximum value of $27.08 \pm 0.30$ kJ/m$^2$. Paired $t$ test indicated that there was a significant difference (*p < 0.05*) in the impact strengths of polyester composites fabricated with untreated and treated sisal and cattail fibers. This difference in impact strengths could be due to the increase in the energy required to break the specimen because of the strong fiber-matrix bonds created by exposing

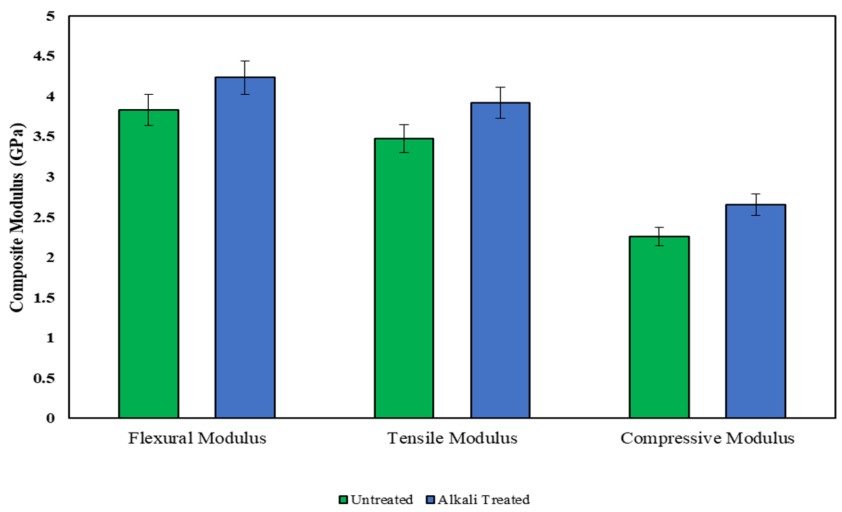

**Figure 4** **Effect of alkali treatment on flexural, tensile and compressive moduli of the hybrid polyester composites.**

hydroxyl groups to the matrix as surface impurities were removed by alkali treatment (*Mortazavi & Moghaddam, 2010*). *Alavudeen et al. (2011)* recorded 24% improvement in impact strength (from 0.50 to 0.62 kJ/m$^2$) for untreated and alkali treated composites at 30 wt% fiber weight fractions for randomly mixed banana/kenaf hybrid polyester composites. Similarly, *Bichang'a, Wambua & Nganyi (2017)* bewrayed that impact strength of a woven sisal/epoxy composite after alkali treatment improved significantly. *Rizal et al. (2018)* also reported that alkali treatment of cattail fibers treated for 1 and 2 h increased the impact strength of epoxy composites from around 10.70 kJ/m$^2$ to 12.40 and 14.20 kJ/m$^2$ respectively. However, treatment for 4 and 8 h recorded a decrease from 14.20 kJ/m$^2$ to 12.80 kJ/m$^2$ in both cases.

It is reported that composites with weak interfacial compatibility may have poor mechanical properties due to crack propagation at the matrix interface (*Chen et al., 2012*). Weak interfacial compatibility in composites also accelerates matrix crack propagation, resulting in debonding of fibers and matrices (*Rizal et al., 2018*). Further, fiber-matrix interface conditions often affect energy absorption in composites, in that composites with good interfacial compatibility always have the impact load received by the polymer matrix transferred to the fiber (*Hao et al., 2018*; *Nair, Wang & Hurley, 2010*) which is its role in matrix reinforcement. Impact failure of composites are caused by fiber and matrix damage, fiber pull-outs from the matrix, and debonding between the fiber and matrix. Debonding is inevitable in the event that the load transferred to the fiber exceeds the fiber-matrix interface strength (*Mylsamy & Rajendran, 2011b*).

## Effects of alkali treatment on thermal conductivity of the hybrid polyester composites

The average thermal conductivity of untreated and treated sisal/cattail hybrid reinforced unsaturated polyester resin composites were $0.66 \pm 0.06$ and $0.72 \pm 0.07$ W/mK respectively

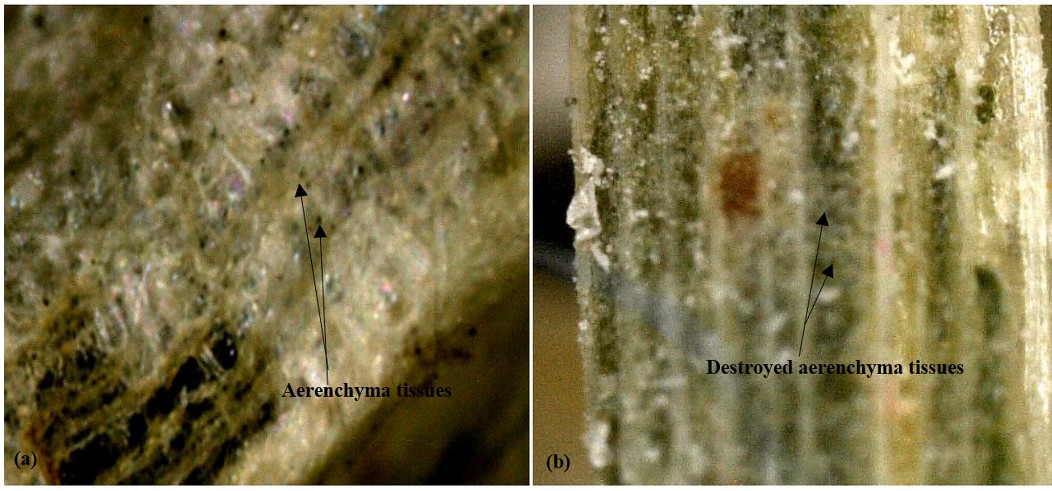

**Figure 5** **Micrographs (x270) showing aerenchyma tissues in cattail fibers.** (A) Untreated; (B) 5% alkali treated.

(Table S2). Thus, thermal conductivity of the composites improved by 8.96% with alkali treatment. However, $t$ test indicated that there was no statistically significant difference ($p < 0.05$) between the thermal conductivities of treated and untreated hybrid composites. This marginal improvement in thermal conductivity with alkali treatment can be attributed to the removal of cementing materials from the fiber surfaces that destroys the aerenchyma tissues within the fibers especially for cattail fibers, reducing the thermal contact resistance of the resultant composites (*Colbers et al., 2017*) (Fig. 5). In addition, alkali treatment improves fiber diameters as well as make fiber surfaces rough. This results in better interlocking between the fibers and the matrix, improving the thermal conductivity of the composites (*Agrawal et al., 1999*; *Zhou, Fan & Chen, 2016*). The recorded thermal conductivity of the composites is close to those previously reported for the individual fibers. Fraunhofer Institute designed a thermal insulation panel made of cattail fibers which had a thermal conductivity of 0.052 W/mK (*Fraunhofer Institute, 2013*). *Luamkanchanaphan, Chotikaprakhan & Jarusombati (2012)* concluded that narrow-leaved cattail fiber (hot-pressed) biocomposite produced with a Methylene Diphenyl Diisocyanate binder had lower thermal conductivities (between 0.0438 to 0.0606 W/mK for density between 200 and 400 kg/m$^3$) and 11–15% moisture diffusion coefficient. *Colbers et al. (2017)* reported thermal conductivity of 1.53 W/mK for blow-in cattail insulation board. Furthermore, *Ramanaiah, Prasad & Reddy (2011)* studying thermal behavior of cattail reinforced polyester composites reported a similar trend with thermal conductivity values between 0.32–0.39 W/mK at a fiber volume fraction between 0.15–0.32. A close thermal conductivity value of 0.16 W/mK at 85% clay was reported by *Dieye, Sambou & Faye (2017)* while investigating the effects of binder (clay) weight on the thermal conductivity of *Typha australis* fiber reinforced composites.

### Fractography studies

Fracture mode in natural fibers can be intracellular or intercellular. The former is often reported in fibers having large elongation that are tested at low speeds. Such a fracture is generally accompanied by tearing of cell walls as well as pull-out of the fibrils (*Mukherjee & Satyanarayana, 1984*). The intercellular fracture is commonly observed in low elongation fibers tested at high speed and occurs typically with separation of the bonding materials between the cells and very little pull-out of the fibrils (*Mukherjee & Satyanarayana, 1984*). In this study, failure mechanism was investigated by inspecting both treated and untreated specimens after the impact test. It was found that the fracture surface of the untreated impact specimens was almost flat while that of treated specimens had saw-like fractured surfaces. Further, more fiber pull-outs were observed in untreated composite specimens, and more fiber breakages in treated specimens (Fig. 3). And thus, the high values of impact strengths reported for treated hybrid composites in this study may also be attributed to their failure modes (larger saw-like fractured surfaces and higher fiber breakages) as more impact energy is absorbed. Therefore, the major composite failure modes identified were fiber pull-outs (in untreated fibers) and fiber fracture (in treated fibers) corroborating a previous observation (*Mbeche, Wambua & Githinji, 2020*). Similarly, *Rizal et al. (2018)* reported that failure in *Typha* fiber reinforced epoxy composites was due to fiber and matrix debonding, fiber pull-outs, and fiber damage.

## CONCLUSION

Alkali treatment of sisal and cattail fibers enhanced the mechanical properties of sisal/cattail fiber reinforced polyester composites. However, there was insignificant enhancement of thermal conductivity of the hybrid composites due to alkali treatment. Fractography studies unveiled that the major failure modes in the resultant composites with untreated and treated fiber blends respectively were fiber pull-outs and fiber fracture. Though the composites produced could be put to non-structural use as ceiling boards, electronic and food packaging materials, further research evaluating properties such as water absorption tendency (wettability), crystallinity, flammability and other thermal properties are inevitable to enhance their efficient practical applications. The effect of extracting sisal and cattail fibers using chelating agents such as sodium tripolyphosphate (STPP) and Ethylenediamine tetraacetic acid (EDTA) on the properties of a similar fiber blend polyester hybrid composite should be investigated.

## ACKNOWLEDGEMENTS

The authors are grateful to the Management of Lomolo Sisal Estate Limited, Mogotio, Kenya and Moi University from which sisal and cattail fibers for this research were sourced. The laboratory services of Moi University Textile Department, Rivatex (East Africa) Limited, Multimedia University, and Jomo Kenyatta University of Agriculture and Technology are highly acknowledged for the analytical success of this research.

### Funding

This research was financially supported by the Africa Centre of Excellence II in Phytochemicals, Textiles and Renewable Energy (ACE II PTRE), Moi University, Uasin Gishu County, Eldoret, Kenya (Credit No. 5798-KE). ACE II PTRE is funded by the World Bank and the Inter-University Council of East Africa (IUCEA). The funders had no role in study design, data collection and analysis, decision to publish, or preparation of the manuscript.

### Grant Disclosures

The following grant information was disclosed by the authors:
Africa Centre of Excellence II in Phytochemicals, Textiles and Renewable Energy (ACE II PTRE), Moi University, Uasin Gishu County, Eldoret, Kenya (Credit No. 5798-KE).
World Bank and the Inter-University Council of East Africa (IUCEA).

### Competing Interests

Timothy Omara is employed by AgroWays Uganda Limited. Silas Mogaka Mbeche declares that there are no competing interests.

### Author Contributions

- Silas Mogaka Mbeche conceived and designed the experiments, performed the experiments, analyzed the data, performed the computation work, prepared figures and/or tables, and approved the final draft.
- Timothy Omara conceived and designed the experiments, analyzed the data, performed the computation work, prepared figures and/or tables, authored or reviewed drafts of the paper, statistical analysis, and approved the final draft.

### Data Availability

The raw measurements are available in the Supplementary Files.

### Supplemental Information

Supplemental information for this article can be found online at http://dx.doi.org/10.7717/peerj-matsci.5#supplemental-information.

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
