# Peer review of "Effects of alkali treatment on the mechanical and thermal properties of sisal/cattail polyester commingled composites"

_PeerJ Materials Science, doi:10.7717/peerj-matsci.5_

## Round 0.1 · original submission · Minor Revisions

Great paper; please address minor revisions suggested by Reviewer 2, after which the paper can be published.

Reviewer 1 ·

Basic reporting

The article is clear and written using professional English language.
The article is quite interesting and I thank you the authors for providing extensive references of similar work done by others. If there is a comment I would like to make is that the general reference listed on line 79 (another study) and line 93 (In an investigation) could be improved by adding either the tittle of the study or investigation on question or just the author involved, despite of the fact that it is well reference (4) and (1) respectively.
I would like to add that I have not found Tables S1 listed on line 221 and Table S2 listed on line 344. The figures are very illustrative and reveal well the findings.

Experimental design

The methodology used is consistent with a comprehensive coverage of the subject.However, I would like to state that the tittle mentions the mechanical and thermal properties, and, only the thermal conductivity was tested. I expected to see more than just one thermal property. Nevertheless, the research is quite meaningful and methods are well described with sufficient information to replicate.

Validity of the findings

I commend the authors for their extensive data set and again comparison with other similar work performed by others. This comparison makes it easier for the reader to understand what to expect from the study. I would like to mention that perhaps there is an error on line 218 in which mentions 26.17 ± 8.33 and 35.17 ± 54.96 tex. I think it might be 26.17 ± 8.33 tex and 35.17 ± 5.496 tex.
Conclusions are well stated and limited to supporting results. Overall the study is well performed, well understood, data have been provided and they are statistically sound.

Reviewer 2 ·

Basic reporting

The manuscript “Effects of alkali treatment on the mechanical and thermal properties of sisal/cattail polyester commingled composites” is a well-written and comprehensive report. The authors provide a good review of relevant literature and compare their own results to those presented in relevant studies.

- The authors should add scale bars to all microscopic images
- In the paragraph “fractography studies” the authors refer to Figure 2 but describe Figure 3. Please change reference to Figure 3.
- The authors are inconsistent with respect to the number of decimals reported for modulus and strength values and should revise accordingly.
- For the caption of Figure 3, I suggest adding a comment that clarifies that fracture surfaces are shown.
- The unit “MPa” is reported as “Mpa” several times, e.g. in lines 283, 287, and 290.

Experimental design

The experiments were designed to thoroughly investigate the posed research question, and the conclusions are supported by data. All methods and experimental procedures are well described.

Validity of the findings

Conclusions are generally supported by data and the authors compare their own findings to results reported in relevant studies. A few things to address:
- The authors discuss that the thermal conductivity does not change significantly between composites with treated and untreated fibers. This contradicts the statement in the conclusion that thermal properties were enhanced for composites contain treated fibers. The authors should change the conclusion so that it matches their findings.
- The impact strength data was not provided in the form of a graph or raw data, numbers are mentioned in the text.

---

## Round 0.2 · accepted · Accept

Your revision has addressed the concerns of the reviewers. Congratulations, the paper has been accepted!